# Differences in self-reported health between cardiac arrest survivors with good cerebral performance and survivors with moderate cerebral disability: a nationwide register study

Karin Larsson  ,[1] Carina Hjelm,[1] Gisela Lilja,[2] Anna Strömberg,[1,3] Kristofer Årestedt[4,5]

[1]Department of Health Medicine and Caring Sciences, Linkoping University, Linkoping, Sweden
[2]Department of Clinical Sciences, Lund University, Lund, Sweden
[3]Department of Cardiology, Linkoping University, Linkoping, Sweden
[4]Linnaeus University Faculty of Health and Life Sciences, Kalmar, Sweden
[5]Department of Research, Region Kalmar County, Kalmar, Sweden

**Correspondence to**
Karin Larsson;
karin.larsson@liu.se

## ABSTRACT

**Objective** The aim was to compare self-reported health between cardiac arrest survivors with good cerebral performance (CPC 1) and survivors with moderate cerebral disability (CPC 2).

**Methods** This comparative register study was based on nationwide data from the Swedish Register of Cardiopulmonary Resuscitation. The study included 2058 in-hospital and out-of-hospital cardiac arrest survivors with good cerebral performance or survivors with moderate cerebral disability, 3–6 months postcardiac arrest. Survivors completed a questionnaire including the Hospital Anxiety and Depression Scale (HADS) and EQ-5D five-levels (EQ-5D-5L). Data were analysed using ordinal and linear regression models.

**Results** For all survivors, the prevalence of anxiety and depression symptoms measured by the HADS was 14% and 13%, respectively. Using the EQ-5D-5L, the cardiac arrest survivors reported most health problems relating to pain/discomfort (57%), followed by anxiety/depression (47%), usual activities (46%), mobility (40%) and self-care (18%). Compared with the survivors with good cerebral performance, survivors with moderate cerebral disability reported significantly higher symptom levels of anxiety and depression measured with HADS, and poorer health in all dimensions of the EQ-5D-5L after adjusting for age, sex, place of cardiac arrest, aetiology and initial rhythm (p<0.001).

**Conclusions** These findings stress the importance of screening for health problems in all cardiac arrest survivors to identify those in need of professional support and rehabilitation, independent on neurological outcome.

## STRENGTHS AND LIMITATIONS OF THIS STUDY

⇒ The study includes a large sample of cardiac arrest survivors (n=2058), both in-hospital and out-of-hospital cardiac arrest survivors with a favourable neurological outcome.
⇒ The data are taken from the Swedish Register of Cardiopulmonary Resuscitation, a nationwide registry with nearly complete coverage of all cardiac arrest in Sweden where cardiopulmonary resuscitation is started.
⇒ The cross-sectional design omits any causal conclusions.
⇒ The reason for drop-outs is unknown and non-response bias can therefore not be excluded.

outcome, commonly assessed with the Cerebral Performance Category (CPC) scale.[4] The CPC classification is made by healthcare professionals on a scale ranging from 1 to 5, where CPC 1 and 2 (ie, good cerebral performance and moderate cerebral disability) are generally defined as a favourable neurological outcome.[5–8] In contrast CPC 3–5 (severe cerebral disability to brain death) are defined as unfavourable neurological outcomes and imply dependency on others.[4 9]

Health-related quality of life (HRQoL) reflects the outcome after a CA from the survivor's perspective and has been recommended as one of the core outcomes post-CA.[10] Health is a broad concept defined by WHO as 'a state of complete physical, mental and social well-being and not merely the absence of disease or infirmity'.[11] Previous studies have shown that CA survivors in general report good health.[12–14] In fact, one comparative study showed that survivors reported even higher levels of health compared with a sex-matched and age-matched normal population.[14] However, a significant minority of the CA

## INTRODUCTION

Cardiac arrest (CA) is a global health problem with poor survival. The survival rate varies across studies and countries but is, according to recent studies, around 20% for in-hospital CA (IHCA)[1] and 10% for out-of-hospital CA (OHCA).[2] CA results rapidly in hypoxic-ischaemic brain damage[3] and the neurological outcome is therefore an important

survivors report severe health problems, particularly with symptoms of anxiety and depression.[12 13 15]

Although cerebral performance and self-reported health are both important outcomes, the association between them is not well explored among CA survivors. A study by Geri *et al*[16] found that survivors with good cerebral performance (CPC 1) reported better health compared with survivors with moderate or severe cerebral disability (CPC 2 and 3). No study has been identified that has specifically investigated differences in self-reported health of CA survivors with good cerebral performance (CPC 1) and moderate cerebral disability (CPC 2). As these survivors are defined as having a favourable neurological outcome, there is a risk that health problems could be overlooked. Therefore, the aim of this study was to compare self-reported health between CA survivors with good cerebral performance (CPC 1) and survivors with moderate cerebral disability (CPC 2).

## METHODS
### Design
This comparative registry study was based on data from the Swedish Register of Cardiopulmonary Resuscitation. Overall, the data are the same as was used in a study by Djärv *et al*[15] that focused on the differences in HRQoL between IHCA and OHCA survivors. However, this study does not include survivors with CPC>2.

### Setting
The Swedish Register of Cardiopulmonary Resuscitation (https://shlrsjh.registercentrum.se/) is a nationwide quality register consisting of two parts: IHCA and OHCA. All CAs are registered when cardiopulmonary resuscitation has been started. The CA survivors are informed of the enrolment in the registry and can withdraw their consent at any time. The register covers almost 100% of OHCA and a majority of IHCA in Sweden. All survivors 18 years or older who are still alive 3–6 months post-CA are screened for eligibility to complete a questionnaire about self-reported health by resuscitation coordinators or cardiac rehabilitation nurses. Unwillingness to participate, severe cognitive dysfunction, language difficulties and severe physical and/or psychological difficulties (according to the latest medical file) are criteria for exclusion in the register during the study period. A letter is sent to the CA survivors, including information about the registry, two questionnaires (the Hospital Anxiety and Depression Scale (HADS) and EQ-5D five-levels (EQ-5D-5L)), and an invitation for a structured interview by phone. The CA survivors are instructed to complete the questionnaires during the structured interview, which also contain an assessment of cerebral performance measured by the CPC score. The interview is performed by the resuscitation coordinators or cardiac rehabilitation nurses. A short-written manual is followed to standardise the assessment of the CPC score.

### Sample and procedure
This study included registry data collected between 1 January 2014 and 31 December 2017. During this period, 2141 CA survivors, 772 OHCA (response rate 54.6%) and 1369 IHCA (response rate 57.4%), had completed the questionnaire. Of these, 57 survivors had CPC≥3 and 26 (1.2%) had missing data on the CPC assessment and were therefore excluded. Thus, the final sample consisted of 2058 survivors.

### Patient and public involvement
The study is based on the Swedish Register of Cardiopulmonary Resuscitation; neither patients nor the public were involved.

### Outcome measures of self-reported health
#### Symptoms of anxiety and depression
The HADS is developed to measure symptoms of anxiety and depression during the last week. The instrument consists of two subscales, HADS anxiety and HADS depression. Each subscale includes seven items, which all have four response options (0–3). The responses to the items in each subscale are summed to a subscale score that can vary between 0 and 21; higher scores indicate higher symptom level.[17] The following cut-off values have been recommended: 0–7 normal range, 8–10 suggested presence of anxiety and/or depression and 11–21 probable presence of anxiety and/or depression.[18] Both subscale scores and cut-off scores were used in this study. The instrument has previously been used in CA research[12 19] and has shown good measurement properties.[20] Cronbach's alpha was 0.88 and 0.85 for the anxiety and depression subscales respectively in this study.

#### Health status
The EQ-5D-5L is a generic instrument to measure health status and includes three parts: EQ-5D-5L descriptive system, EQ Visual Analogue Scale (VAS) and EQ-5D value index (the latter not used in this study).[21] The EQ-5D-5L descriptive system consists of five health dimensions: mobility, self-care, usual activities, pain/discomfort and anxiety/depression. Each dimension has five response options, referred to as levels, ranging from no problems (1) unable to do/extreme problems (5). These scores reflect the intensity of the health problems. The five levels can also be dichotomised into no problem (1) and problems (2–5), reflecting the prevalence of the health problems. Both prevalence and intensity scoring are used in this study. The EQ VAS is a measure of overall health and graded on a 0–100 scale, where 0 represents worst possible health and 100 best possible health.[21] The EQ-5D-5L has been recommended as a core outcome in CA research[10] and has in previous studies shown good measurement properties in different patient groups, for example, stroke and chronic obstructive pulmonary disease.[22 23]

## Statistical analysis

To present the characteristics of the survivors and the variables, descriptive statistics were used. To compare self-reported health between CA survivors with good cerebral performance (CPC 1) and survivors with moderate cerebral disability (CPC 2), $\chi^2$ test was used for nominal data and Mann-Whitney U test was used for ordinal data.

Multiple linear regression analyses were used to investigate associations between cerebral function (CPC as the explanatory variable) and self-reported health (the EQ VAS and HADS as outcome variables). Multiple ordinal logistic regression analyses were used to handle the ordinal nature of data in the EQ-5D-5L health dimensions. Cerebral function (CPC 1=0, CPC 2=1) was included as an explanatory variable and the EQ-5D-5L health dimensions as outcome variables. Both the linear and ordered logistic regression models were adjusted for covariates that are commonly used in CA research[12 15]; age, sex (male=0, female=1), place of CA (IHCA=0, OHCA=1), aetiology (heart disease=1, other=0), and initial rhythm. Initial rhythm was coded as shockable rhythm (ventricular fibrillation (VF) /ventricular tachycardia (VT)) and non-shockable rhythm (pulseless electrical activity (PEA) and asystole). To handle the large number of missing data in initial rhythm (n=341), they were coded within a third 'unknown' group. Thus, initial rhythm was included as a dummy coded variable in the regression models, with shockable rhythm as a reference category.

The level of statistical significance was set at p<0.05 and the data were analysed using SPSS statistics V.24 (IBM) and Stata V.16.1 (StataCorp).

## RESULTS
### Survivor characteristics

The final sample consisted of 2058 survivors: 1309 IHCA survivors and 749 OHCA survivors. The median age was 68 years (q1-q3=59–75), and the majority were male (n=1438, 70%). Most survivors were reported to have good cerebral performance (CPC 1, n=1838, 89%). A majority of the CA were witnessed (n=1902, 94%) and caused by heart disease (n=1472, 73%). About three out of four survivors had an initial shockable rhythm (n=1242, 72%), and 63% (n=1297) were defibrillated. The median time to defibrillation was 2min (q1-q3=1–8), 1min for IHCA and 8min for OHCA. Compared with survivors with moderate cerebral disability, those with good cerebral performance were significantly younger (p=0.002), more often male (p=0.009), had more often an initial shockable rhythm (p<0.001), were more often defibrillated (p=0.002) and more often had heart disease as the cause of CA (p<0.001). In addition, the share of OHCA was larger in the group of survivors with good cerebral performance than in the group of survivors with moderate cerebral disability (p=0.011). The difference in time to defibrillation and whether the CA was witnessed did not show any significant difference between survivors with good cerebral performance and survivors with moderate cerebral disability (table 1).

### Symptoms of anxiety and depression (HADS)

Subscale scores for the HADS showed that the survivors generally reported low symptom levels of anxiety (Mdn=2, q1-q3=0–5) and depression (Mdn=2, q1-q3=1–5). However, survivors with moderate cerebral disability reported significantly higher symptom levels of both anxiety (Mdn=4 vs. 2, p<0.001) and depression (Mdn=5 vs. 2, p<0.001) than survivors with good cerebral performance (table 2). Based on the multiple linear regression analyses, these differences remained when age, sex, place of CA, aetiology and initial rhythm were adjusted for (p<0.001). The regression model explained 8% of the total variance in symptoms of anxiety and 7% in symptoms of depression (table 3).

Using the cut-off scores for the HADS, 280 (14%) of all the CA survivors reported having suggested or probable presence of anxiety and 261 (13%) suggested or probable presence of depression. However, the share of survivors with suggested or probable presence of anxiety and depression was significantly higher among survivors with moderate cerebral disability than in survivors with good cerebral performance (p<0.001). Among the survivors with moderate cerebral disability, 53 (24%) reported having symptoms of anxiety and 64 (29%) symptoms of depression (table 4).

### Health status (EQ-5D-5L and EQ VAS)

Using the EQ-5D-5L descriptive systems, the CA survivors reported pain/discomfort as the most prevalent health problem (n=1172, 56.9%), followed by anxiety/depression (n=971, 47.2%), usual activities (n=952, 46.3%), mobility (n=822, 40.0%) and self-care (n=365, 17.7%). All health problems were significantly more prevalent among survivors with moderate cerebral disability than survivors with good cerebral performance (p<0.001) (figure 1).

Although health problems were common, the CA survivors reported, on average, low intensity levels of all health problems (Mdn=1–2). However, the survivors with moderate cerebral disability reported significantly higher intensity levels for all health dimensions compared with the survivors with good cerebral performance (p<0.001). Survivors with good cerebral performance reported the highest levels of pain/discomfort (Mdn=2, q1–q3=1–2) while survivors with moderate cerebral disability reported the highest levels of usual activities (Mdn=3, q1–q3=2–4) (table 2). Based on the ordinal regression analyses, these group differences remained when age, sex, place of CA, aetiology and initial rhythm were included as adjusting covariates (p<0.001). The pseudo $R^2$ values (McFadden) ranged between 0.02 and 0.08 (table 5).

The CA survivors reported overall a median score of 75 (q1–q3=60–85) on the EQ VAS; survivors with moderate cerebral disability reported significantly worse health status than survivors with good cerebral performance (Mdn=60 vs. 75, p<0.001) (table 2). Based on the multiple linear regression analysis, this group difference remained in the adjusted model (p<0.001). The regression model explained 10% of the total variance in EQ VAS (table 3).

**Table 1** Background data of the survivors with good cerebral performance (CPC 1) and survivors with moderate cerebral disability (CPC 2)

| | Valid n | All n=2058 | CPC 1 n=1838 | CPC 2 n=220 | P value |
|---|---|---|---|---|---|
| Age (years), Mdn (q1–q3) | 2057 | 68 (59–75) | 68 (58–75) | 71 (62–79) | 0.002* |
| Sex, n (%) | 2058 | | | | 0.009† |
| Male | | 1438 (69.9) | 1301 (70.8) | 137 (62.3) | |
| Female | | 620 (30.1) | 537 (29.2) | 83 (37.7) | |
| Witnessed CA, n (%) | 2034 | | | | 0.071† |
| Yes | | 1902 (93.5) | 1691 (93.2) | 211 (96.3) | |
| No | | 132 (6.5) | 124 (6.8) | 8 (3.7) | |
| Initial rhythm, n (%) | 1717 | | | | <0.001† |
| Shockable | | 1242 (72.3) | 1142 (74.3) | 100 (55.6) | |
| Non-shockable | | 475 (27.7) | 395 (25.7) | 80 (44.4) | |
| Time to defibrillation (min), Mdn (q1–q3) | 1173 | 2 (1–8) | 2 (1–8) | 2 (1–7) | 0.544* |
| Place, n (%) | 2058 | | | | 0.011† |
| IHCA | | 1309 (63.6) | 1152 (62.7) | 157 (71.4) | |
| OHCA | | 749 (36.4) | 686 (37.3) | 63 (28.6) | |
| Defibrillation, n (%) | 2031 | | | | 0.002† |
| Yes | | 1297 (63.9) | 1181 (65.0) | 116 (54.2) | |
| No | | 734 (36.1) | 636 (35.0) | 98 (45.8) | |
| Aetiology, n (%) | 2026 | | | | <0.001† |
| Heart disease | | 1472 (72.7) | 1339 (74.1) | 133 (61.0) | † |
| Other | | 554 (27.3) | 469 (25.9) | 85 (39.0) | |

*Mann-Whitney U test.
†$\chi^2$ test.
CPC, Cerebral Performance Category; IHCA, in-hospital cardiac arrest; OHCA, out-of-hospital cardiac arrest.

## DISCUSSION

To the best of our knowledge, this is the largest study to have compared self-reported health between CA survivors with good cerebral performance (CPC 1) and survivors with moderate cerebral disability (CPC 2). Although these groups are commonly defined as having a favourable neurological outcome after their CA, the results show that survivors with moderate cerebral disability reported significantly more health problems, measured by the HADS and EQ-5D-5L, compared with survivors with good

**Table 2** Differences in self-reported health between survivors with good cerebral performance (CPC 1) and survivors with moderate cerebral disability (CPC 2)

| | | All n=2058 | CPC 1 n=1838 | CPC 2 n=220 | P value* |
|---|---|---|---|---|---|
| EQ-5D-5L | Mobility, Mdn (q1–q3)† | 1 (1–2) | 1 (1–2) | 2 (1–3) | <0.001 |
| | Self-care, Mdn (q1–q3)† | 1 (1–1) | 1 (1–1) | 1 (1–2) | <0.001 |
| | Usual activities, Mdn (q1–q3) | 1 (1–2) | 1 (1–2) | 3 (2–4) | <0.001 |
| | Pain/discomfort, Mdn (q1–q3) | 2 (1–3) | 2 (1–2) | 2 (1–3) | <0.001 |
| | Anxiety/depression, Mdn (q1–q3)† | 1 (1–2) | 1 (1–2) | 2 (1–3) | <0.001 |
| EQ VAS | EQ VAS, Mdn (q1–q3) | 75 (60–85) | 75 (60–85) | 60 (45–70) | <0.001 |
| HADS | Anxiety, Mdn (q1–q3) | 2 (0–5) | 2 (0–5) | 4 (1–7) | <0.001 |
| | Depression, Mdn (q1–q3) | 2 (1–5) | 2 (0–4) | 5 (3–8) | <0.001 |

*Mann-Whitney U test.
†Based on 2057 observations.
CPC, Cerebral Performance Category; EQ-5D-5L, EQ-5D descriptive system five-levels; HADS, Hospital Anxiety and Depression Scale; VAS, Visual Analogue Scale.

**Table 3** Associations between cerebral function (CPC) and self-reported health (EQ VAS and HADS); multiple linear regression analyses adjusted for age, sex, place of cardiac arrest, aetiology and initial rhythm (n=2025)

| Outcome variables | Explanatory variables | B (SE) | 95% CI for B | P value |
|---|---|---|---|---|
| EQ VAS | CPC 2 | −15.02 (1.39) | −17.75/−12.29 | <0.001 |
| | Model statistics | $F_{(7, 2,017)}=31.8$, $p<0.001$, $R^2=0.10$ | | |
| HADS anxiety | CPC 2 | 1.86 (0.27) | 1.33/2.38 | <0.001 |
| | Model statistics | $F_{(7, 2,017)}=23.3$, $p<0.001$, $R^2=0.08$ | | |
| HADS depression | CPC 2 | 2.61 (0.25) | 2.12/3.12 | <0.001 |
| | Model statistics | $F_{(7, 2,017)}=20.7$, $p<0.001$, $R^2=0.07$ | | |

The adjusting covariates are excluded from the table.
CPC, Cerebral Performance Category; EQ-5D-5L, EQ-5D descriptive system five-levels; HADS, Hospital Anxiety and Depression Scale; VAS, Visual Analogue Scale.

cerebral performance. These differences remained after adjustment for age, sex, place of CA, aetiology and initial rhythm.

The survivors generally reported low symptom levels of anxiety and depression, but a great difference was shown between those with moderate cerebral disability and those with good cerebral performance. The prevalence of anxiety and depression in this study corresponds well with a systematic review about psychological health post-CA.[24] Most of the included studies that had used the HADS reported a prevalence of 13%–14% of both anxiety and depression, thus levels were similar to those shown for the whole sample in this study. It should be noted that the prevalence of anxiety and depression in this study was significantly higher in survivors with moderate cerebral disability. The prevalence of anxiety in this group corresponds with a Swedish study of 278 OHCA survivors treated with target temperature management, at 6 months' follow-up.[25] However, the prevalence of depression in that study was lower compared with the survivors with moderate cerebral disability in this study. This finding was unexpected as all patients in the temperature study, in contrast to this study, had received intensive care and thereby experienced a more complicated disease trajectory.

The prevalence of depression was more common than anxiety in survivors with moderate cerebral disability and the opposite situation was shown for survivors with good cerebral performance. Included studies in a systematic review by Wilder Schaaf et al reported higher prevalence of anxiety than depression, or equal levels, in CA survivors. However, the opposite situation was not reported. It should be noted that only OHCA survivors were included in that review, the number of included studies was small (n=11), and the sample size in all included studies was small (21–168 survivors). In addition, different measures were used to assess symptoms of anxiety and depression, and only four studies used the HADS.[24] The HADS' ability to differentiate between anxiety and depression has been questioned, and a review about the psychometric properties concluded that the subscales therefore should be compared with caution.[26]

Unfortunately, there are no Swedish normative data for symptoms of anxiety and depression measured by the HADS. Compared with normative data for anxiety and depression measured by the HADS in the UK,[27] the prevalence of anxiety was less common while depression was more common in survivors with good cerebral performance. In addition, the prevalence of having suggested or probable presence of anxiety and depression was more

**Table 4** Differences between survivors with good cerebral performance (CPC 1) and survivors with moderate cerebral disability (CPC 2) in HADS cut-off score

| | All n=2058 | CPC 1 n=1838 | CPC 2 n=220 | P value |
|---|---|---|---|---|
| Anxiety, n (%) | | | | <0.001 |
| Normal range (0–7) | 1778 (86.4) | 1611 (87.6) | 167 (75.9) | |
| Suggested presence (8–10) | 165 (8.0) | 140 (7.6) | 25 (11.4) | |
| Probable presence (11–21) | 115 (5.6) | 87 (4.7) | 28 (12.7) | |
| Depression, n (%) | | | | <0.001 |
| Normal range (0–7) | 1797 (87.3) | 1641 (89.3) | 156 (70.9) | |
| Suggested presence (8–10) | 159 (7.7) | 117 (6.4) | 42 (19.1) | |
| Probable presence (11–21) | 102 (5.0) | 80 (4.4) | 22 (10.0) | |

CPC, Cerebral Performance Category; HADS, Hospital Anxiety and Depression Scale.

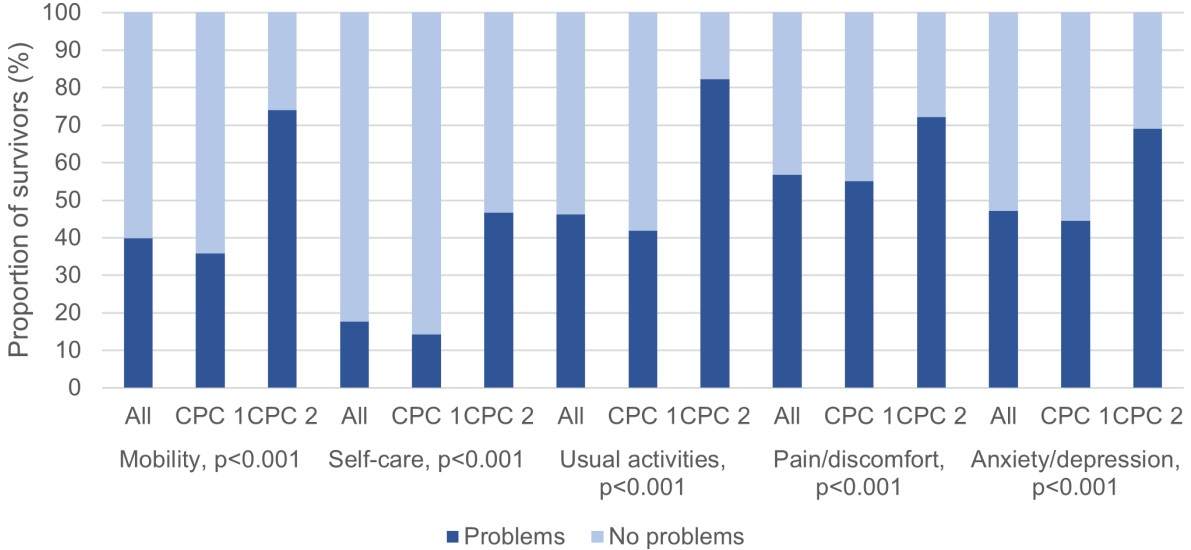

**Figure 1** Prevalence scoring of the EQ-5D-5L health dimensions, dichotomised into no problems (1) and problems (2–5), in relation to neurological outcome (good cerebral performance=CPC 1 and moderate cerebral disability=CPC 2). CPC, Cerebral Performance Category.

common in survivors with moderate cerebral disability compared with normative data from the UK. It should be noted that normative data from the UK may differ from the general population in Sweden. However, the results indicate that symptoms of anxiety and depression need to be acknowledged also in CA survivors with favourable neurological outcome.

Survivors with moderate cerebral disability also reported more problems with their physical and overall health, measured with the EQ-5D-5L, compared with survivors with good cerebral performance. Age and comorbidity may explain these differences as survivors with moderate cerebral disability were significantly older and more often suffered IHCA compared with survivors with good cerebral performance. However, the multiple regression analyses in this study show that age could not explain these

differences. Unfortunately, data of comorbidity is not registered for OHCA survivors in the Swedish Register of Cardiopulmonary Resuscitation and could therefore not be examined in this study. It is likely that comorbidity is an important confounder as previous studies have shown that comorbidity is related to self-reported health in CA survivors[28] and is likely to be related to cerebral performance as well. Comorbidity should therefore be considered in future research.

Compared with a Swedish population-based study, including 25 867 adults between 30 and 103 years,[29] the CA survivors with good cerebral performance reported equal median scores on the EQ VAS as persons with low education level, the group with poorest health status in the general population. However, survivors with moderate cerebral disability reported in average 15 points lower on

**Table 5** Association between cerebral function (CPC) and self-reported health (EQ-5D-5L); multiple ordinal logistic regression analyses adjusted for age, sex, place of cardiac arrest, aetiology and initial rhythm (n=2024–2025)

| Outcome variables | Explanatory variables | OR (SE) | 95% CI for OR | P value |
|---|---|---|---|---|
| Mobility | CPC 2 | 3.75 (0.50) | 2.89/4.86 | <0.001 |
| | Model statistics | LR $\chi^2$(7, n=2024)=350.7, p<0.001, McFadden $R^2$=0.08 | | |
| Self-care | CPC 2 | 4.30 (0.64) | 3.21/5.77 | <0.001 |
| | Model statistics | LR $\chi^2$(7, n=2024)=202.8, p<0.001, McFadden $R^2$=0.08 | | |
| Usual activities | CPC 2 | 4.88 (0.65) | 3.76/6.33 | <0.001 |
| | Model statistics | LR $\chi^2$(7, n=2025)=222.7, p<0.001, McFadden $R^2$=0.05 | | |
| Pain/discomfort | CPC 2 | 2.00 (0.27) | 1.54/2.60 | <0.001 |
| | Model statistics | LR $\chi^2$(7, n=2025)=107.9, p<0.001, McFadden $R^2$=0.02 | | |
| Anxiety/depression | CPC 2 | 3.02 (0.42) | 2.31/3.96 | <0.001 |
| | Model statistics | LR $\chi^2$(7, n=2024)=144.7, p<0.001, McFadden $R^2$=0.03 | | |

The adjusting covariates are excluded from the table.
CPC, Cerebral Performance Category; EQ-5D-5L, EQ-5D descriptive system five-levels.

the EQ VAS compared with persons with low education level in the general population. Moreover, the presence of health problems in the five health domains in EQ-5D-5L were similar between the survivors with good cerebral performance and the general population. In contrast, survivors with moderate cerebral disability reported more health problems in all health dimensions in EQ-5D-5L than persons with low education in the general population. The CA survivors in this study also reported lower median scores on EQ VAS compared with survivors who had been treated with therapeutic hypothermia 6 months earlier.[30] Thus, problems with self-reported health needs to be taken seriously also in CA survivors with favourable neurological outcome.

## Methodological limitations

The study has some limitations that need to be considered. No causal relationship can be drawn due to the cross-sectional design. The no-response rate was acceptable; however, the reasons for not participating are not known. It is a potential risk that the response rate is lower among survivors with moderate cerebral disability and those with poor self-reported health. In addition, survivors with severe physical and/or psychological difficulties were excluded from the third registration. Therefore, we cannot be sure that the participants are representative of the entire population of CA survivors. Information about survivors' cerebral performance before CA is unknown and therefore, we do not know if the impaired cerebral performance among those survivors who were classified according to CPC 2 was caused by the CA. The use of CPC may be criticised as it did not take to account prior disability. Of this reason, other measures like the Modified Ranking Scale and extended Glasgow Outcome Scale are commonly recommended to be used in CA survivors. However, the correlation between these outcome measures is high.[10 31] To exclude the risk that the assessment of the CPC at the follow-up assessment would be affected by the survivors' self-reported health during the interview, the agreement with CPC at hospital discharge was examined. The absolute agreement was 81% and the risk for that is therefore rather small. The HADS and EQ-5D-5L were used to measure self-reported health as they are recommended in international guidelines for post CA care.[10 32] However, neither HADS nor EQ-5D-5L are disease specific for CA survivors and might therefore not capture specific health problems related to surviving a life-threatening condition such as CA. Even so, the instruments capture important health problems, and early screening could be a way to identify survivors at risk of low self-reported health, also in survivors with favourable neurological outcome, particularly in survivors with moderate cerebral disability.

## CONCLUSIONS

This study contributes with important knowledge about cerebral performance and self-reported health in CA survivors with favourable neurological outcome. Survivors with moderate cerebral disability (CPC 2) reported significantly lower levels of self-reported health compared with survivors with good cerebral performance (CPC 1). These findings stress the importance of screening for health problems in all CA survivors to identify those in need of professional support and rehabilitation, independent of neurological outcome. Until there are CA-specific instruments to measure self-reported health, the HADS and EQ-5D-5L serve as adequate screening instruments.

**Acknowledgements** We would like to thank all the participating survivors in the Swedish Register of Cardiopulmonary Resuscitation.

**Contributors** KL: acting as guarantor and contributed to the conception of the design, the acquisition of the data, analysis, and interpretation of data, drafting and revising the work critically for important intellectual content and approval of the final manuscript. CH: contribute to the conception of the design, interpretation of data, drafting and revising the work critically for important intellectual content and approval of the final manuscript. GL: contributed to the conception of the design, revising the work critically for important intellectual content and approval of the final manuscript. AS: contributed to the conception of the design, revising the work critically for important intellectual content and approval of the final manuscript. KÅ: contributed to the conception of the design, the acquisition of the data, analysis, and interpretation of data, drafting and revising the work critically for important intellectual content and approval of the final manuscript.

**Funding** This work was supported by The Medical Research Council of Southeast Sweden (FORSS-745341 and FORSS-939845) and Region Östergötland, Linköping University (RÖ-920281), Sweden.

**Competing interests** None declared.

**Patient and public involvement** Patients and/or the public were not involved in the design, or conduct, or reporting, or dissemination plans of this research.

**Patient consent for publication** Not applicable.

**Ethics approval** This study involves human participants and was approved by the Regional Ethical Review Board in Linköping, Sweden No. 2018/480-21. The CA survivors have been informed of the enrolment in the registry and could withdraw their consent at any time.

**Provenance and peer review** Not commissioned; externally peer reviewed.

**Data availability statement** Data are available on request from the author KL.

**ORCID iD**
Karin Larsson http://orcid.org/0000-0002-8376-705X

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
