## [Reviewer comments · BMJ Open]

ARTICLE DETAILS

TITLE (PROVISIONAL)	Differences in self-reported health between cardiac arrest survivors with good cerebral performance and survivors with moderate cerebral disability – a nationwide register study
AUTHORS	Larsson, Karin; Hjelm, Carina; Lilja, Gisela; Stromberg, Anna; Årestedt, Kristofer

VERSION 1 – REVIEW

REVIEWER	Darrigo, Sonia Fondazione Policlinico Universitario Agostino Gemelli IRCCS
REVIEW RETURNED	09-Dec-2021

GENERAL COMMENTS	Thank you for giving me the opportunity to review this manuscript that I read with interest. The study compare self-reported health of cardiac arrest survivors with good cerebral performance (CPC 1) and survivors with moderate cerebral disability (CPC 2). The strength of study is the large sample size of survivors with CPC 1-2. The burden of mental health disorders is high among survivors of cardiac arrest. There is an urgent need to understand the predisposing risk factors and develop preventive strategies, in addition to identify patients that need an early professional support and rehabilitation. The manuscript is well written and clear; the rationale is clearly stated, these results may help to know what is the rate of neurological disorder in patients with good outcome. I have no comment regarding methods and results. Limitations are perfectly handled. I have no important concern and I propose it for publication.
--

REVIEWER	Tiainen, Marjaana Helsinki University Central Hospital
REVIEW RETURNED	25-Jan-2022

GENERAL COMMENTS	Reviewer comments to manuscript "Self-reported health in cardiac arrest survivors with favourable neurological outcome - a nationwide comparative register study" Authors: Karin Larsson, Carina Hjelm, Gisela Lilja, Anna Strömberg and Kristofer Årestedt Manuscript ID bmjopen-2021-058945 Manuscript Type: Original research This is an observational study based on a prospective registry (Swedish register on cardiopulmonary resuscitation). The aim of the study was to compare self-reported health in cardiac arrest survivors
---

with CPC scores 1 or 2 – both scores are usually considered to represent good neurological outcome after cardiac arrest with functional independence. The instruments used for self-reported health were EQ-5D-5L and Hospital Anxiety and Depression Scale, and the data was collected 3-6 months after cardiac arrest by questionnaires, after a telephone interview. The patient cohort is large, 2058 survivors, of whom 1838 had CPC 1 and 220 CPC 2 outcome.

In general, 14% of survivors with favourable neurological outcome reported symptoms of anxiety and 13% reported symptoms of depression. Some health problems were often reported, but these were in general mild. The authors found that survivors with moderate cerebral disability (CPC 2) reported significantly higher levels of anxiety and depression, and poorer health in all five dimensions in EQ-5D-5L and also VAS, compared to those survivors with CPC 1 outcome. The differences persisted after statistically adjusting for confounding factors.

The manuscript is well written, and the aim of the study is clear. As the authors state, this is the first study to compare self-reported health parameters within cardiac arrest survivors with CPC 1 or CPC 2 outcomes. Although these outcomes are considered as good, this study shows that especially in CPC2 survivors experience health problems and report worse health-related quality of life.

Comments to the authors

Methods

1. Page 6, first line: a structured interview is mentioned. Could you briefly describe here what this interview includes? Questions about ADL-functions, need of help, mobility, activities outside home, return to work etc?

2. Page 6, first chapter: The interview is performed by the resuscitation coordinators or cardiac rehabilitation nurses. At the time of interview, cerebral performance is also assessed based on the interview, using the CPC score.

Are there some training or written guidelines for the evaluators on how to assess CPC? Something comparable to training for evaluating modified Rankin Scale?

3. Page 7, Health status: Why was EQ-5D value index not used?

4. Are there Swedish normative data for EQ-5D-5L? If this exists, it would be very interesting to compare the health-related quality of life of cardiac arrest survivors to that of general age- and sex-matched Swedish population (VAS or EQ value index). In Discussion it is mentioned that there are no Swedish normative data for HADS, so unfortunately this comparison is not possible.

Results

5. Page 9, first line: The median time to defibrillation was 2 minutes. If I understand correctly, this is counted from collapse? The very short time-interval to defibrillation then reflects the fact that about 2/3 of the study patients had in-hospital cardiac arrest?

Discussion

6. Discussion is mainly about findings on anxiety and depression measured by HADS, and in my opinion this is justified, because the results are novel and in part unexpected. However, I suggest that you would add some comparison of EQ VAS in this study compared to previously reported EQ VAS in cardiac arrest survivors.

VERSION 1 – AUTHOR RESPONSE

Reviewer 1:	Response:
	Thank you for your comments.
1-The study compare self-reported health of cardiac arrest survivors with good cerebral performance (CPC 1) and survivors with moderate cerebral disability (CPC 2). The strength of study is the large sample size of survivors with CPC 1-2.	
2-The burden of mental health disorders is high among survivors of cardiac arrest. There is an urgent need to understand the predisposing risk factors and develop preventive strategies, in addition to identify patients that need an early professional support and rehabilitation.	
3-The manuscript is well written and clear; the rationale is clearly stated, these results may help to know what is the rate of neurological disorder in patients with good outcome.	
4-I have no comment regarding methods and results.	
5-Limitations are perfectly handled.	
6-I have no important concern and I propose it for publication.	
Reviewer 2:	Response:
1-Methods Page 6, first line: a structured interview is mentioned. Could you briefly describe here what this interview includes? Questions about ADL-functions, need of help, mobility, activities outside home, return to work etc?	The structured interview is conducted to collect data from the self-reported measures and to assess cerebral performance by the CPC score by phone. We realise that the wording was misleading and have revised the text to make this clearer.
2-Methods Page 6, first chapter: The interview is performed by the resuscitation coordinators or cardiac rehabilitation nurses. At the time of interview, cerebral performance is also assessed based on the interview, using the CPC score. Are there some training or written guidelines for the evaluators on how to assess CPC? Something comparable to training for evaluating modified Rankin Scale?	The resuscitation coordinators or cardiac rehabilitation nurses who perform the interviews are following a written manual when they score the survivor's CPC based on the interview. Additionally, they do not have a specific training to assess CPC. The resuscitation coordinators and/or cardiac rehabilitation nurses are annually invited to a meeting to discuss the registry and registrations such as how to score the CPC. This to ensure the quality of the registration of the CPC and to strengthen the validity of how the questions are asked. We have added information about the written

	manual to the method section.
3- Methods Page 7, Health status: Why was EQ-5D value index not used?	The reason for not including the EQ-5D value index is that this study focusses on patient reported outcome measures. The EQ-5D value index is a preference-based measure and is developed for health economic evaluations. No revision has therefore been made.
4- Methods Are there Swedish normative data for EQ-5D-5L? If this exists, it would be very interesting to compare the health-related quality of life of cardiac arrest survivors to that of general age- and sex-matched Swedish population (VAS or EQ value index). In Discussion it is mentioned that there are no Swedish normative data for HADS, so unfortunately this comparison is not possible.	This is an important comment, thank you. We have compared the findings with general population study using EQ VAS in the discussion section.
5-Results Page 9, first line: The median time to defibrillation was 2 minutes. If I understand correctly, this is counted from collapse? The very short time-interval to defibrillation then reflects the fact that about 2/3 of the study patients had in-hospital cardiac arrest?	This is true that the time to defibrillation was short. This may be understood by the fact that 63% was in-hospital CA and a great majority (>90%) of both IHCA and OHCA were witnessed (which is highly expected since we only have those who have survived the CA). For your information, the median time was 1 minute for IHCA and 8 minutes for OHCA. These times are now reported in "Survivor characteristics".
6-Discussion Discussion is mainly about findings on anxiety and depression measured by HADS, and in my opinion this is justified, because the results are novel and in part unexpected. However, I suggest that you would add some comparison of EQ VAS in this study compared to previously reported EQ VAS in cardiac arrest survivors.	We agree with this remark. As a response to this comment and the comment about normative data above, we have added a discussion about our results in comparison to health status in a general population and in CA survivors.

VERSION 2 – REVIEW

REVIEWER	Tiainen, Marjaana Helsinki University Central Hospital
REVIEW RETURNED	10-Jun-2022
GENERAL COMMENTS	I thank the authors for their detailed answers to my questions and comments, and for the additions and clarifications made in the manuscript. I have no further questions or comments.